# Age-Dependent Redistribution of the Life-Important Enzyme in the Retina: Adult Müller Glial Cells’ Endfeet Lack Spermine Synthase Expression

**DOI:** 10.3390/biom15101374

**Published:** 2025-09-27

**Authors:** Astrid Zayas-Santiago, Christian J. Malpica-Nieves, José M. Santiago, Yanitza Hernández, David E. Rivera-Aponte, Miguel Méndez-González, Rüdiger W. Veh, Legier V. Rojas, Serguei N. Skatchkov

**Affiliations:** 1Department of Pathology and Laboratory Medicine, Universidad Central del Caribe, Bayamón, PR 00956, USA; astrid.zayas@uccaribe.edu; 2School of Chiropractic, Universidad Central del Caribe, Bayamón, PR 00956, USA; 3Department of Physiology, Universidad Central del Caribe, Bayamón, PR 00956, USA; legier.rojas@uccaribe.edu; 4Department of Natural Sciences, University of Puerto Rico-Carolina, Carolina, PR 00984, USA; jose.santiago13@upr.edu; 5Department of Biochemistry, Universidad Central del Caribe, Bayamón, PR 00956, USA; yanitza.hernandez@uccaribe.edu (Y.H.); david.rivera@uccaribe.edu (D.E.R.-A.); miguel.mendez3@upr.edu (M.M.-G.); 6Department of Natural Sciences, University of Puerto Rico, Aguadilla, PR 00603, USA; 7Institut für Zell- und Neurobiologie, Charité–Universitätsmedizin Berlin, Centrum 2, Charitéplatz 1, D-10117 Berlin, Germany; ruediger.veh@charite.de

**Keywords:** polyamines, spermine, spermine synthase, nervous system, retina, glial Müller cells, neuronal synapses, glial cell compartments, aging

## Abstract

Polyamine (PA) spermine (SPM) (i) plays an essential role in the function of neurons, while (ii) accumulating predominantly in glial cells by an unknown mechanism. In addition, the translocation of SPM synthesis and redistribution in the developing and maturating retinas remains unclear. Therefore, the expression of the SPM-synthesizing enzyme, spermine synthase (SpmS), was compared in rat retinas on postnatal days 3, 21, and 120 using immunocytochemistry, Western blot (WB), and ImageJ analyses. The anti-glutamine synthetase (GS) antibody identified glial cells, and DAPI labeled the cell nuclei. At postnatal day 3 (P3), the neonatal retina shows widespread SpmS expression throughout most neuroblast cells, but absent in the developing synaptic layers and Müller cell (MCs) processes. By day 21 (P20), SpmS becomes strongly expressed in neurons, and not in glia. On day 120 (P120), SpmS was observed in synaptic areas, with significantly less presence in neuronal soma and still none in MCs. WBs showed a decrease in SpmS expression during maturation. Therefore, glial cells do not synthesize SPM, and the accumulation of SPM in MCs found earlier suggests that glial cells take up SPM via a hypothetical high-affinity SPM transporter. In glia, SPM regulates glial connexin (Cx43) and potassium (Kir4.1) channels, being a key player in CNS diseases and aging.

## 1. Introduction

Polyamine (PA) SPM, a biological organic polycation, is a key metabolic molecule without which life is impossible [1,2,3]. SPM is synthesized from putrescine or agmatine via spermidine (SPD) [4,5], and it is necessary to emphasize that SPM and SPD are localized in the glial cells of the whole brain [6] and the retina, but not in neurons [7,8,9]. These facts can suggest a key role of glia and PAs in providing new insights into PA exchange between glia and neurons in the central nervous system (CNS). Both deficits (i) of SPM/SPD [4,5,6,10,11,12,13,14,15,16] and (ii) glial cells (that accumulate PAs) [17] are critical for CNS (patho)physiology because SPM regulates a large number of channels, receptors, and transporters [1,2,3,4,5,11,12,13,14,15,16,17,18].

PAs are key players in (i) CNS diseases [16,19,20,21,22,23], (ii) synaptogenesis [24,25], (iii) astrocyte survival, and (iv) gliogenesis [26]. PAs promote the regeneration of retinal neurons after trauma [27], enhance longevity [16,28,29], boost memory [24,30], and trigger autophagy [21,22,28]. Recent data suggest that fasting therapy leads to SPD accumulation, which induces autophagy and hypusination of the translation regulator eIF5A, ultimately resulting in increased longevity [31]. However, neither the role of glial cells nor the localization of the spermine synthase (SpmS), a spermine-synthesizing enzyme, has been analyzed in glia.

On the other hand, a mutation in the SpmS gene results in a loss of function and a lack of SPM production in Snyder–Robinson syndrome (SRS), leading to intellectual, muscular, and metabolic disabilities, as well as early death [1,2,32,33]. SpmS deficiency causes deafness and extreme sensitivity to a blocker of ornithine decarboxylase, the enzyme that synthesizes the primary polyamine putrescine [34]. Another example is Gyro mice, which lack SPM due to an X-chromosomal deletion affecting the SpmS gene. These mice experience sterility, deafness, neurological abnormalities, reduced body size, and sudden death [1]. Intriguingly, knockdown of SpmS in Drosophila induces SRS-like syndrome in homozygotes; however, partial SpmS deficiency in heterozygotes is associated with increased lifespan [23]. This suggests that the presence of SpmS is necessary to maintain the SPD/SPM ratio. Indeed, partial knockdown of SpmS reduces Tau protein accumulation, which is a promising method for studying Alzheimer’s disease [23,35].

Alternatively, when SPM content declines via the degradation pathway, the catabolizing enzyme spermine oxidase (SMOX) produces dangerous molecules from SPM, such as acrolein, other aldehydes, and hydrogen peroxide, toxic radicals. This toxicity leads to severe (i) neuronal damage, (ii) reactive gliosis in astrocytes, and (iii) severe brain trauma [5,36,37]. Therefore, several problems were highlighted due to (i) a deficit of PAs in Alzheimer’s disease [23,35], (ii) changes in SPM/SPD ratio as an indicator of sarcopenia during aging [38], and (iii) the common toxicity of SPM degradation products [2,5,39,40,41]. It may be, therefore, suggested that the accumulation of SPM/SPD in glia [6,7,8] is probably (i) a protective mechanism to clean up SPM from the extracellular space, and (ii) to avoid oxidation with the following toxicity of released radicals (iii), and to control SPM concentration in the CNS [11,18].

Free concentration of SPM^+4^ cations may reach 1 mM in glial retinal Müller cells [42], where SPM opens Cx43 gap junctions to help astrocytes [43,44] and MCs [45] to contact each other in their syncytium. Recently, astrocytic syncytium was described as isopotential [46], which may be due to the opening of connexin gap junctions (Cx GJs) [43,44]. These bring new insights to study SPD/SPM exchange in glia because PAs have antihyperalgesic, antioxidative, antiapoptotic, antifibrotic, antinociceptive, anticonvulsant, and antiseizure effects [5,22,47]. PAs also have antidepressant [12], neuroprotective [27], anti-inflammatory, and memory-facilitating effects [30], as well as being helpful against Multiple Sclerosis, Alzheimer’s disease, diabetes-2, etc. [18]. Acetylated PAs (a-PAs) are markers of several disorders of the brain. Acetylated SPD is accumulated in and released from astrocytes and is a hallmark of HIV-associated neurological disorder (HAND) [48]. Acetylated putrescine is a hallmark of Parkinson’s disease [49]. Therefore, the study of PAs and a-PAs accumulated in glia suggests that these molecules are gliotransmitters regulating synaptic plasticity [11,25,47]. PA-associated brain function is an emerging avenue.

SPD [9] and SPM [7,8] are primarily located in the endfeet of adult Müller glial cells. Since SPM is synthesized from SPD, the following remains unclear: (i) Is the synthesis of SPM from SPD by SpmS present in the glial endfeet of cells in the adult CNS, particularly in the retina? and (ii) Why do SPD and SPM accumulate in astrocytes [6] and retinal glial cells? [7,8,9]. Consequently, given that radiolabeled SPM [50] or biotinylated SPM (b-SPM) [13] typically do not permeate from outside the CNS through the blood–brain barrier (BBB), the accumulation of SPM in glial cells appears to be solely due to internal mechanisms within the CNS [13]. The synthesis of the SPM precursor SPD by spermidine synthase has been previously observed in adult retinal neurons [51] and in adult brain neurons [52]; however, SPD has also been detected in MCs [9]. Furthermore, the reason for the increase in retinal SPM and SPD levels in the course of early development, followed by a decline by postnatal day 16, remains uncertain [53]. Considering that SPM regulates numerous glial and neuronal receptors, age-related variations in PA synthesis and concentration may significantly impact retinal function. This work aims to determine whether SPM accumulation in glial cells is solely due to the presence of SpmS in glia or if the accumulation of SPM [7,8] in glia results from an unidentified PA-uptake transporter. Potential PA uptake has been identified in various cell types via (i) broad-specificity and low-affinity organic cation transporters (OCTs) named SLC22A-1,-2,-3 [54] and SLC18B1 [55]; (ii) a high-affinity transporter more specific to SPM, such as ATP13A2/PARK9 [19,56]; and (iii) large pores like Cx43 hemichannels, which may offer an alternative pathway [57]. However, the production of SPM within glial cells has not yet been reported. Therefore, we studied the localization and age-related movement of SpmS in the retina.

## 2. Materials and Methods

### 2.1. Animals and Tissues

Rats were housed in a 12 h light-dark cycle room in a standard cage and had access to standard water and food freely. Sprague Dawley rats (3, 21, and 120 days old) were sacrificed, and eyes were extracted and prepared for Western blot and immunohistochemical analyses as described [51].

All experiments were conducted following the ARVO Statement on the Use of Animals in Ophthalmic and Vision Research and in compliance with NIH requirements. All procedures received approval from the Universidad Central del Caribe Institutional Animal Care and Use Committee under protocol #018-2024-01-00, adhering to the National Institutes of Health guidelines for the humane treatment of laboratory animals.

### 2.2. Immunohistochemistry

Sprague Dawley rats’ eyes were enucleated and fixed for 45 min in two different solutions. First solution for spermine synthase labeling consisted of 4% paraformaldehyde (Sigma-Aldrich, P6148, St. Louis, MO, USA) in phosphate-buffered solution 0.1 M (PBS: NaCl 136.9 mM, Na_2_HPO_4_ 10.1 mM, KCl 2.7 mM, KH_2_PO_4_ 1.8 mM with pH 7.4). In addition, if needed for labeling SPM/SPD accumulation, we added to the above-described solution 0.2% Picric Acid and 0.05% Glutaraldehyde [51]. Eyes were penetrated in the ora serrata with a 25-G needle, and fixed with fresh fixative solution for an additional 20 min. The eyes were rinsed three times with 0.1 M PBS (phosphate-buffered saline), opened, and the retinas were removed in a cold PBS solution using a stereoscopic microscope (Fisher Stereomaster, FW00-20-1613, Waltham, MA, USA). After fixation, retinae were embedded in 4% Agarose (Gibco BRL, 15510-019, Big Cabin, OK, USA) in PBS 0.1 M. Leica VT 1000 S Vibratome (Leica, Wetzlar, Germany) was used to create 20 μm retinal sections.

Tissue samples were transferred to a 24-well plate, and cell membranes were permeabilized for 20 min with a 1% DMSO (MP Biomedicals, 02196055, Santa Ana, CA, USA) and 0.3% Triton X-100 (Sigma, T9284) in a 0.1 M PBS. After removing the permeabilization solution, samples were treated with a blocking solution for one hour, which contained 0.3% Triton X-100 (Sigma, T9284) in 0.1 M PBS mixed with 2% bovine serum albumin (BSA; Sigma, A4503), and either 5% normal goat serum (Vector, S-1000) or 5% normal horse serum (Vector, S-2000, Burlingame, CA, USA), along with 1% DMSO (MP, 196055), depending on the secondary antibody. The blocking solution was removed and replaced with the primary antibodies, and left at 4 °C overnight in the shaking box.

Different primary antibodies for the detection of (i) SpmS and (ii) GS were used. Unconjugated (i) rabbit polyclonal anti-spermine synthase (Abcam, ab248996, dilution 1:250) to determine SpmS localization and (ii) glutamine synthetase (GS) (mouse anti-glutamine synthetase (Millipore, Burlington, MA, USA MABN1182, dilution 1:250)) to visualize Müller glial cell anatomy and co-localization of the labels. The antibodies for SpmS and GS exhibited strong immunoreactivity without cross-reactivity. According to a vendor, Abcam, the anti-SpmS antibody (rabbit, monoclonal, ab248996, 41 kDa) does not exhibit cross-reactivity with the anti-spermidine synthase (SpdS) antibody. Previously, we utilized the anti-SpdS antibody (rabbit, polyclonal, ab111884, 34 kDa), which also shows no cross-reactivity with SpmS. After incubating the retinal sections, the primary antibody aliquot was removed, and a permeabilization solution was used to wash the samples for 10 min and three times while shaking. Red fluorescence anti-mouse Texas Red (Vector, Tl-2000, diluted 1:200) and Green fluorescence anti-rabbit FITC (Vector, Fl-1000, diluted 1:200) were prepared in the permeabilization solution and added to the samples. The 24-well plate with samples was then covered to shield it from light (avoiding bleaching of fluorescent tags), and incubated for two hours at 4 °C with constant shaking. Following incubation, the samples were washed three times for 10 min with PBS 0.1 M and once with distilled water.

Tissue samples were mounted on slides and allowed to dry for 5 min. After drying, either Fluoroshield with DAPI (Sigma, F6057) or Hard Set Vectashield with DAPI (Vector, H-1500) was applied before sealing the slides with coverslips. Consequently, all retinal samples were triple-labeled with GS, a specific marker for Müller glial cells; SpmS, a marker for spermine synthesis; and DAPI, which stains nuclei and retinal layers.

### 2.3. Confocal Microscopy

Confocal images were acquired using an Olympus BX60 microscope (Olympus, Tokyo, Japan), which was equipped with the Olympus FV1000 confocal laser scanning system. Images were captured at 40-x magnification. To ensure reliability, all experiments were conducted in triplicate. All images were processed using the ImageJ (NIH), (Version 1.52u, NIH, Bethesda, MD, USA) maintaining consistent confocal parameters across all captured images. Adobe Photoshop was utilized to create the figures.

### 2.4. Western Blot

The tissue homogenization buffer was prepared at pH 7.5 and contained the following components (in mM): 20 Tris–HCl, 150 NaCl, 1.0 EDTA, 1.0 EGTA, 1.0 phenylmethylsulfonyl fluoride (PMSF), 1% Triton X-100, and a mixture of protease inhibitors, including leupeptin, bestatin, pepstatin, and aprotinin. Proteins were separated on a 10% polyacrylamide-SDS gel (20–80 µg of protein/lane). Electroblotting was performed using the trans-blot turbo transfer system (cat #1704150, Bio-Rad, Hercules, CA, USA) onto a nitrocellulose membrane. The membrane blocking was performed using the Odyssey Blocking Buffer (cat #927-40000, LI-COR Biosciences, Lincoln, NE, USA) at room temperature for 1 h. The primary antibodies used for immunodetection of the proteins in this study were as follows: an anti-rabbit SpmS antibody (1:5000 dilution, cat #ab248996, Abcam, Cambridge Science Park, Cambridge, UK), which was incubated in blocking solution overnight at 4 °C. For the secondary antibodies, goat anti-rabbit (1:25,000, cat #926-32211, LI-COR Biosciences, Lincoln, NE, USA) was used for 1 h at room temperature.

As our loading control, we applied an anti-mouse β-actin antibody (1:5000, cat #A5441, Sigma-Aldrich, St. Louis, MO, USA). This was followed by incubation with the secondary goat anti-mouse antibody (1:25,000, cat #926-32210, LI-COR Biosciences, Lincoln, NE, USA) for 1 h at room temperature.

Infrared signals from the membranes were detected using the LI-COR Odyssey model CLx Scanner (LI-COR Biosciences, Lincoln, NE, USA). The membranes were stained with India ink to determine total protein, allowing for quantification of small differences in sample loading. Final detection was conducted using enhanced chemiluminescence (SuperSignal^®^ West Dura Extended Duration Substrate; Rockford, IL, USA), as per the manufacturer’s instructions. The signal was quantified with a gel documentation system (ChemiDoc, Bio-Rad, Hercules, CA, USA), and images were captured using Image Lab software (Version 6.1, Bio-Rad, Hercules, CA, USA). Original figures can be found in Appendix A. 

### 2.5. Quantitative Image Analysis of Fluorescent Staining

All images were analyzed using ImageJ software (version 2.1.0/1.53c). ImageJ is a powerful, open-source image processing and analysis tool developed by the National Institutes of Health. This software is widely used in research, particularly in the life sciences, for tasks such as image visualization, measurement, and analysis. It is available in several distributions, including the core ImageJ and the Fiji distribution, which is known for its emphasis on biological image analysis. We measured fluorescence in seven spots across four different regions/layers of the retina: the ganglion cell layer (GCL), neuroblast layer (NBL), inner plexiform layer (IPL), and outer plexiform layer (OPL) in each retinal sample image (*n* = 3). The SpmS fluorescence images were obtained through confocal microscopy and analyzed using the ImageJ software (Version 1.52u, NIH, Bethesda, MD, USA) [26,51]. We measured the fluorescence in each spot and calculated the mean fluorescence, which was then imported into PRISM (Version 9.4.1 (458), GraphPad Software, San Diego, CA, USA) for statistical analysis. 

### 2.6. Data Analysis and Statistics

Two-way ANOVA with Tukey’s multiple comparison test was utilized to compare the mean fluorescence of the SpmS samples. Additionally, one-way ANOVA with Tukey’s multiple comparison test was employed to compare the mean fluorescence of the other samples. Statistical significance was determined at *p*-values lower than 0.05, corresponding to a 95% confidence interval. 

## 3. Results

### 3.1. Immunocytochemistry for Glutamine Synthetase in Glia, Spermine Synthase in Retinal Cells, and DAPI in Cell Nuclei of Rat Retina at Different Ages

We analyzed Spermine Synthase (SpmS, Figure 1, green label) expression in the rat retinae from the earlier developmental neonatal period (3 days postnatal: P3, Figure 1), in the maturating period (21 days postnatal: P20, Figure 2), and in adulthood (120 days postnatal: P120, Figure 3). The localization of this biosynthetic enzyme producing spermine was identified and compared with other immunolabels in the retina. Consequently, we used an antibody of the glial marker Glutamine Synthetase (GS, red label) to identify Müller glial cells and a marker of cell nuclei DAPI (blue label).

At the P3 (early postnatal stage), when retinal cells are still immature, DAPI staining (blue, Figure 1, left panel) reveals two primary cellular layers: GCL, containing newly formed ganglion cells, and the NBL, which houses the nuclei of developing neurons, including amacrine, bipolar, horizontal, and rods and cones (photoreceptor cells).

Between the two layers, GCL and NBL, lies the space occupied by synapses known as the IPL (white horizontal arrows), which lacks cell bodies, and there is no DAPI staining in the IPL where the synaptic zone is forming. At this stage, NBL represents a mixture of nuclei from most neurons that will mature later, and the INL has not yet formed in the NBL. Furthermore, the outer nuclear layer (ONL) and the OPL are also not yet formed, as clearly visible in Figure 2, P20.

On the other hand, spermine synthase, SpmS (green color, Figure 1), is already expressed in the early postnatal retina. In (i) the area of newborn ganglion cells and (ii) the neuroblast layer, young neurons are developing and differentiating from neuroglial progenitors within the neuroblast layer (NBL). The expression of spermine synthase (SpmS) is present throughout the NBL and in the retinal tissue, particularly within the ganglion cell layer. The glial cell marker glutamine synthetase (GS, indicated by a red label) is found in all layers of the retina, signifying the presence of radial glia. In the NBL, the nuclei of newly formed neurons that produce spermine are observed migrating along the radial glial fibers (Figure 1, indicated by yellow arrows).

At the early age (P3), robust expression of glutamine synthetase (GS) was observed (indicated by red labeling in Figure 1) throughout all layers of the retina, but specifically in the terminal processes of glial cells, the endfeet, and the distal cell part (Figure 1, blue and yellow arrows). GS serves as an essential marker for glial cells, particularly highlighting healthy radial glia. We chose not to use GFAP, an alternative glial marker, because it identifies reactive rather than healthy resting glial cells. GS is responsible for synthesizing glutamine. Glutamine is released from glia and taken up by neurons for the production of the neurotransmitter glutamate. In P3 rat retinas, GS was distributed across the entire retina (Figure 1, red), reflecting the predominance of glial progenitor cells at this developmental stage. Additionally, glial cells can be seen enveloping all nuclei within the NBL.

When merged (right panel, Figure 1), SpmS expression is seen in all types of neuronal cells but not in Müller cell processes at this early stage of postnatal retina. Blue arrows indicate the glial cell processes that form membranes known as the inner (Figure 1e) and outer (Figure 1f) limiting membranes. GS labels at the edges of the retina (blue arrows) represent glial Müller cell processes. Intriguingly, SpmS labels are completely absent in these glial Müller cell processes. Additionally, SpmS is not expressed in the forming synapses of the inner plexiform layer (white arrows), which differs from what is observed at age P20 and P120 (see below, Figure 2). At this developmental stage, the retinal cells within the neuroblast layer are primarily progenitor cells that have not yet fully differentiated into either neuronal or glial lineages. Identifying the specific types of neurons is beyond the scope of the present study.

Since the immunostaining was performed on thick tissue (retinal slides) instead of a thin monolayer of cells (as in cell culture), the overlap of several cells in certain areas represents a simple physical layer-by-layer stack. Therefore, it indicates not the co-localization of the immunolabels but merely an overlap of different colors of fluorescence. However, it is clear that in the terminal processes of glial cells, where no neurons are present, such as the endfeet and distal processes (blue arrows, Figure 1), no overlaps have been observed, and there is no SpmS labeling in these processes (Figure 1, Figure 2 and Figure 3). Thus, the only region where glial cells are present but neurons are absent is the area of glial cells’ endfoot processes. In this region, SpmS expression is absent across all ages tested so far.

During the maturation retinal age, P20 rats (Figure 2), SpmS is most strongly expressed in ganglion cells, where ganglion cell bodies are surrounded by Müller cell processes (Figure 2A, merge), less in the bipolar cells (in the INL), and strongly in the first synaptic layer, outer plexiform layer (in the OPL), between bipolar cells and photoreceptor cells (Figure 2A). GS immunoreactivity was not present in neurons. At this age, all cells are fully differentiated [20]. GS was clearly presented only in Müller cells and their compartments, such as the endfeet, stalks, somata, and distal processes. GS shows strong expression on the inner nuclear layer, where the bipolar cell, horizontal cell, and Müller cell somata are located (INL), on the outer plexiform layer where synapses of photoreceptors are located (OPL) and very weakly on the outer nuclear layer of inner segments of cones and rods, (the photoreceptor cells (ONL)), the synaptic area between photoreceptors and bipolar cells. Radial glial Müller cells extend all retinal layers up to the rods’ and cones’ outer segments (Figure 2A, GS, and Merge). The endfeet of Müller cells make a border between the retina and vitreal space called the inner limiting membrane (ILM), where GS labels are pronounced.

In P120 rat retinas (Figure 3), SpmS exhibits its highest expression in the neuronal synapses and terminals within the IPL (Figure 3, green). Only faint labeling appears in the ganglion cells of the GCL. Neurons in the INL demonstrate little to no SpmS staining, with expression confined to their synapses and terminals. When images are merged, SpmS expression does not overlap with that of glutamine synthetase (GS) (Figure 3, right panel).

At this age, the neurons are fully mature. GS-immunoreactivity was not observed in neuronal cells, but it was significantly expressed only in all Müller cell compartments, including the endfeet, somata, proximal processes (stalks), and distal processes, with strong expression through the INL (where the Müller cell somata are located) and through the ONL (the layer of inner segments of photoreceptor cells (cones and rods). Müller cells extend across the entire thickness of the retina, reaching up to the outer segments of rod and cone photoreceptors. In P120 rats, GS labeling indicates fully developed radial glial cells. No label of SpmS was found in the neuronal soma of bipolar cells in INL, nor was any label of SpmS found in the nuclei of photoreceptors in ONL (pink arrows, Figure 3). However, there is a strong expression of SpmS in the outer segments of photoreceptors. Again, practically no merging staining of SpmS and GS was observed in both terminal processes of Müller cells, such as endfeet and distal processes of these glial radial cells. This demonstrates that glial cells do not express SpmS.

As previously reported [9], in P120 rats, spermidine primarily accumulates in the proximal processes, including endfeet and stalks of Müller cells, particularly near the bodies of ganglion cells and within endfeet. These endfeet contribute to the formation of the inner limiting membrane (ILM) [7,8,9,18]. The ILM consists of Müller cell endfeet that interface with the vitreous space of the retina, an area where GS labeling is highly expressed (Figure 1, Figure 2 and Figure 3), yet SpmS is absent from this region. At this stage, polyamine buildup specifically occurs in these Müller cell compartments [7,8,9] that lack SpmS, indicating the accumulation of spermidine [9] and spermine [7,8] without local synthesis. Importantly, earlier studies demonstrated minimal polyamine labeling in the ONL, which includes the inner segments of photoreceptors, in adult P120 rats [9], as well as in the retinas of guinea pigs [7] and frogs [8]. Since the localization of SPD/SPM is exclusively found in Müller cells across many species, including humans, with the subsequent absence of this PA accumulation in neurons [7,8,9], we may suggest that the synthesis of either SPD or SPM occurs in some neurons, but PA accumulation is necessary in glial cells via transporter systems. The candidates may include SLC18B [55], ATP13A4, or SLC22A1-,2-,3 [54] or other systems such as small and large pores and channels [5,11,13,44].

### 3.2. Western Blot Analysis for SpmS Content in Rat Retina of Different Ages

Western blot analysis was performed to assess variations in SpmS concentration across the entire retinal tissue. The amount of SpmS in the whole retina decreases with age after day 21 (Figure 4). We normalized the data to 100% for retinae from P3 rats. SpmS protein content shows an increase to about 150% by day P20 but declines dramatically in P120 rats. During early adulthood, from days 21 to 120, SpmS activity drops significantly; however, SpdS activity and content remain stable during this period, as was shown recently [51].

The β-actin was used as the loading control, for normalization of the data, and to justify stable conditions for all measurements of SpmS content.

### 3.3. SpmS Redistribution Between Retinal Layers During Aging

SpmS distribution in each retinal layer was also evaluated using the semi-quantitative image analysis, as previously applied to spermidine synthase expression [51] and biotinylated-SPM uptake experiments [26], which support WB data. Confocal microscopy images showing SpmS fluorescence were analyzed using ImageJ software (version 2.1.0/1.53c). This approach facilitates a comparison of fluorescence signal intensity. Quantitative fluorescence measurements were performed in four distinct retinal regions across samples from different developmental stages. Specifically, for SpmS analysis, fluorescence intensity was recorded from seven spots in each of the following retinal areas: (i) the ganglion cell layer (GCL) at the endfoot zone; (ii) the inner plexiform layer (IPL), which contains ganglion cell synapses; (iii) the neuroblast layer, comprising progenitor cells; and (iv) the outer plexiform layer, where photoreceptor synapses are located. Each measurement was conducted on images from retinal samples (*n* = 3) across all selected spots (Figure 5). It is evident that in P3 rats, the SpmS signal (Figure 5, 3 days) is strongest in the GCL and NBL but nearly absent in the IPL (synaptic layer, black columns). In the maturing retinae, SpmS signal intensities in the GCL (P20, red columns) are slightly higher than at P3, while in the IPL-synaptic layer, SpmS fluorescence is significantly higher than at P3. Additionally, in P20, a new layer has formed, the OPL, which displays strong SpmS fluorescence. Notably, in P120 rats, this difference became particularly pronounced (represented by gray columns). First, the ganglion cells exhibited a drop in SpmS signals of almost half compared to P3 and P20. Second, IPL synaptic SpmS activity increased about threefold compared to P20. Third, the OPL synaptic SpmS signals also show progressive growth.

## 4. Discussion

Functionally, spermine, the product of SpmS, regulates various receptors, channels, and transporters [2,3,4,5,7,8,9,10,11,18]. In glial cells, SPM influences the behavior of the Kir4.1 channels [7,8,18,20]. Here we show that SpmS is not expressed in the adult Müller glial cells, and, therefore, SPM is not produced in these cells, which is clearly visible from P20 and P120 ages (Figure 2 and Figure 3). Therefore, understanding the localization of PA synthesis and accumulation in the CNS [5,11,13,18] is important because many receptors found in neurons and glia are sensitive to SPM. PAs block imidazoline and NMDA, AMPA glutamate receptors [58], and since NMDA and GluA2-lacking AMPA receptors are calcium-permeable, this SPM blockade of these receptors protects neurons from calcium-induced overload and apoptosis. PAs also regulate acetylcholine (AChR) receptor channels [59], Kir channels [42,60], TRP channels [61,62], Cx43 channels [63], GluA-6 kainate receptors [64], ASIC channels [65,66], and many others.

In glial cells specifically, (1) SPM controls the rectification of the Kir4.1 channels [42], (2) removes the blocking effect of calcium and hydrogen cations on glial Cx43 gap junctions (GJs) [63,67], (3) enhances astrocyte-to-astrocyte coupling [43], and (4) selectively transports negatively charged organic molecules via GJs in the glial syncytium [44]. In addition to potassium buffering, SPM-sensitive Kir4.1 channels are essential for glutamate clearance [68] and are extremely important for proper CNS function [69]. The KCNJ10 gene, which encodes the potassium channels Kir4.1 (the major channel found in glial cells), is expressed between postnatal days 11 and 20, while in humans, they are expressed from approximately 2 to 6 years after birth [20]. After this period, gene expression in the healthy CNS remains relatively stable in both humans and rodents [20]. Intriguingly, we found that SpmS is not expressed at this age.

In rodents, P3 approximately corresponds to 6 months in humans, while P20 equates to about 4 to 6 years in humans [20]. The P120 pertains to a mature adult. Thus, the accumulation of SPM in glial cells is not the result of their synthesis but rather due to transport mechanisms. Consequently, exploring the polyamine mystery, studying PA exchange and function is critically important [5,10,12,18,22,70,71].

The retina can be viewed as a simply layered, accessible model of the brain, as it ontogenetically arises from a common origin, the neural tube [72]. Previous research has shown that SPM precursor, SPD, is produced by spermidine synthase (SpdS) in adult neurons, specifically in ganglion cells of the rat retina, but not in glial cells [51]. However, SPD is translocated into glial cells by uptake from neurons rather than synthesis in Müller cells [9]. Similarly, SPM was identified not in neurons but in Müller glial cells across different species, including humans [7,8]. The antibodies used to detect SpdS [51] and SpmS [current studies] are non-cross-reactive antibodies according to Abcam. Here, we demonstrate that (i) SpmS is limited to adult neurons and not present in glia, and (ii) SpmS activity may change in ganglion cells and synapses with aging (Figure 1, Figure 2, Figure 3, Figure 4 and Figure 5). This is an important finding, as recent attention has focused on SpmS activity in neurons concerning Alzheimer’s-type changes in the CNS. Tao and co-authors [23] found that the reduction in SpmS enhances autophagy, thereby suppressing Tau accumulation in neurons.

In detail, we investigated the behavior of SpmS in the retina to identify the primary sources of spermine production across different stages of aging. Our findings reveal several important insights: first, SpmS was detected in neuronal progenitor cells within the neuroblast layer during early phases of cell proliferation and differentiation (P3, Figure 1), but there was no evidence of SPM synthesis in Müller cell endfeet (Figure 1, blue arrows). As retinal maturation progressed (P20, Figure 2), SpmS labeling remained absent in glial cells, and at the aging stage (P120), Müller cells clearly lacked SpmS expression (Figure 3). Second, because SpmS consistently co-localized with neurons (DAPI marker) but not with the glial marker (GS marker) in Müller cells during both the maturation (P20) and early aging (P120) stages, we infer that neurons are the primary source of SPM. Third, with advancing age, SpmS was predominantly expressed in ganglion cells, and only rarely in other neuronal types (Figure 3). A novel finding is the strong SpmS expression in the synaptic regions of the mature retina (Figure 2, Figure 3 and Figure 5).

In summary, since SPM accumulation was previously identified in cell compartments known as endfeet of Müller cells that encase ganglion cells [7,8,9], this is evidently not due to synthesis but rather a local translocation of SPM from synapses and neuronal bodies to glial processes and somata. Such polyamine accumulation in Müller cells’ endfoot processes (where SpmS is absent (Figure 1, Figure 2 and Figure 3) has been observed across various animal species and in humans [7,8,9,18], indicating a conserved, evolutionarily stable accumulation of SPM/SPD within these glial cells by transporting systems. Notably, cultured astrocytes did not survive or proliferate without supplementation of exogenous spermidine (SPD) [26]. This indicates that glial cells depend on the external uptake of PAs, and the protective effect of SPD on developing glial cells was negated when PA transport was hindered using a specific transporter inhibitor [26].

Our findings indicate that Müller cells play a fundamental role in polyamine exchange. We propose that, despite lacking the ability to synthesize spermidine (SPD) [51] and spermine (SPM) (Figure 1, Figure 2, Figure 3, Figure 4 and Figure 5), Müller cells can retain these PAs and potentially serve as donors to release PAs back to neurons during critical periods, thereby contributing to neuroprotection through glial SPD/SPM release. Conversely, in conditions such as injury or disease where glial PA accumulation is impaired, aging neurons may be unable to produce sufficient levels of SPD and SPM on their own. As a result, external PA supplementation becomes essential for the survival of retinal neurons [27].

Glial Müller cells are known to accumulate spermidine and spermine to high levels [7,8,9], reaching concentrations of free SPM^4+^ cations in adult glial cells nearly 1 mM [42]. In contrast, events such as mechanical trauma of the brain [73], ischemic conditions [74], and exposure to mono-fluoroacetate, the gliotoxin [47] trigger substantial or complete and rapid release of SPM/SPD from glial cells [47]. Future research should focus on elucidating the mechanisms of PA uptake and release by glial cells, particularly in older animal models and models of retinal disease

Polyamine catabolism through degradation [14,36,37,40,47,70,74], oxidation, and the generation of toxic byproducts like acrolein [39,40] has been linked to several CNS disorders and diseases, such as Parkinson’s [75,76,77], Alzheimer’s [78,79], Huntington’s diseases [30], and syndromes such as Snyder-Robinson, EAST/SeSAME, Down, and Rett [18,20,80]. On the other hand, intact (non-catabolized) polyamines can: (i) protect against oxidative stress [2,41,81]; (ii) reduce glutamate-induced toxicity [70]; (iii) prevent DNA damage [40,82]; (iv) enhance memory function [83,84]; and (v) contribute to extended lifespan [22,24,25,28,29]. Additionally, polyamines are essential for cone photoreceptor maturation in the retina [53], and SpmS expression is evident in the outer segments (OS) of photoreceptors, as shown in Figure 3.

In the retina, polyamines (PAs) modulate Kir4.1 channels in Müller glial cells [7,8,20,42,80] and can permeate these channels [68]. PAs also activate glial Cx43 channels [43,44,45], which are expressed in retinal Müller cells [45], and can restore their function by preventing blockage from hydrogen [67] and calcium ions [63]. Furthermore, SPM may pass through Cx43 channels [43,44,45,63,67], facilitating the transport of other molecules via this pathway. Consequently, PAs play crucial roles in multiple central CNS processes [12,21,25,83,85,86] and help to reduce oxidative stress in animal models of optic nerve-related multiple sclerosis [87]. Inhibition of PA oxidases also has therapeutic potential with amino-guanidine (a PA oxidase inhibitor), which aids in the regeneration of facial nerves after injury in rats [88].

Polyamine content restoration in the brain and retina helps in diseases, trauma, and aging [25,27,83,89,90]. Interestingly, a daily intake of SPD reduces retinal ganglion cell death and promotes optic nerve regeneration after injury [27], while PA deficiency disrupts the migration and survival of retinal pigment epithelial cells [91]. There is an age-related decline in the levels of SPD and SPM content in neurons both in the brain [52,92] and the retina [9]. Despite the absence of SPD and SPM synthesis in adult glial cells [51] current study, these glial cells are still the primary contributors to the presence of SPD and SPM.

Figure 6 summarizes our findings on the localization of SpmS and SpdS [51], and SPM/SPD in adult retina (Figure 6A), suggesting polyamine exchange between neurons, glia (Figure 6B), and potential pathways for SPD/SPM in glial cells (Figure 6C). Recently, a novel high-affinity spermine transporter, ATP13A4 (Figure 6C [93]), was identified in glial cells, which clarifies the essential uptake of polyamines at low SPM concentrations outside of these cells. Astrocytes communicate with each other [46,94] and SPM, SPD open their Cx43 gap junctions [63,67], and enhance cell-to-cell signaling [44,45]. Such Cx43 GJs form a glial syncytium and are referred to as an isopotential syncytium [46,94] (Figure 6B, Cx43 Gap). Alternative pathways for SPM/SPD entrance into astrocytes could be the half-gap junctions, called the hemichannels. Therefore, such accumulation of PAs in glia is necessary to open connexin-43 (Cx43) gap junctions in astrocytes [44], in retinal Müller cells [45], and dietary spermidine may protect against retinal disease [95]. Intriguingly, photons from external illumination pass via the endfoot, the stalk of Müller cells to rods and cones [18,96], and all such pathways are filled with polyamines that probably serve for such functions [97].

## 5. Conclusions

Recent emergent data demonstrate that PAs are key players in HIV associated neurocognitive disorder (HAND); Alzheimer’s, Parkinson’s, and Huntington’s diseases; several syndromes; and aging. PAs accumulate in glial cells, and it was shown that PAs are released from glia by glial-specific treatments.

Together with an age-dependent decline of SPD and SPM content in neurons in the brain and retina, the most important players that hold SPD and SPM remain glial cells, regardless of the lack of SPD synthesis and SPM synthesis [current study] in such adult glia. We suggest further research to identify a high-affinity polyamine transporter in glial cells, as the previously identified SLC22A3 transporter exhibits a low affinity for taking up sub-micromolar concentrations of polyamines.

Indeed, a novel high-affinity spermine transporter, ATP13A4, was recently identified in glia, which explains the essential uptake of PA at low SPM concentrations outside glia. Since PAs open gap junctions, this enlarges cell-to-cell communication and creates a glial isopotential syncitium, and, in addition, SPM and SPD may potentially participate in new functions of glial cells, such as light guidance found earlier in Müller cells. Therefore, we consider that PAs are novel “gliomediators” that open Cx43 GJs and keep glial syncytium integrity and isopotentiality and may help to maintain healthy brain status. Furthermore, astrocytes convert PAs partially to GABA, which is released to regulate synaptic activity, suggesting a unique internal PA exchange in the CNS. Since PA content declines with aging, the neuroprotection afforded by PAs can decline as well, which substantially increases the risk of morbidity and mortality. Notably, exogenous and dietary spermidine exhibits neuroprotective properties against DNA damage, oxidative stress, neurotoxicity, cognitive decline, retinal diseases, and brain injury, while also extending lifespan.

## Figures and Tables

**Figure 1 biomolecules-15-01374-f001:**
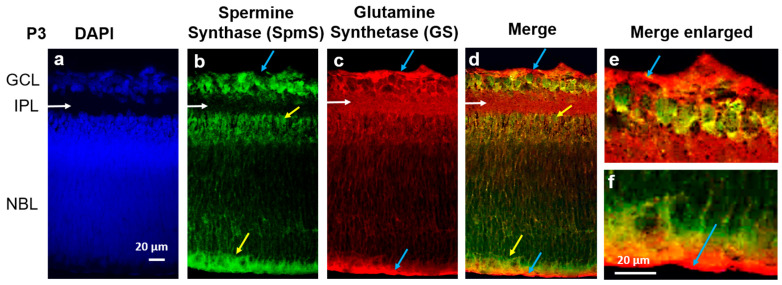
Localization of Spermine Synthase (SpmS) in retinal P3 progenitors. (**a**) Nuclei marker DAPI (blue), (**b**) Spermine Synthase (SpmS, green), (**c**) glial cell marker Glutamine Synthetase (GS, red), in 3-day-old (newborn: P3) rat retina. DAPI labels all cell nuclei, while SpmS is localized in the nuclei of progenitors, including newborn neurons migrating along glial fibers (yellow arrows) in the IPL (white arrows), and in the NBL (yellow arrows). It is also clear that glial cells (labeled by GS, red labeling) surround all cell nuclei in NBL since the glial marker GS is widely expressed in radial Müller cells through all the layers. The SpmS expression in progenitors is poorly co-localized with GS as the glial marker, and no co-localization is visible in the endfeet of glial Müller cells that form the inner limiting membrane (blue arrows, (**d**,**e**)) and in distal processes of Müller cells that form the outer limiting membrane (blue arrows, (**d**,**f**)). SpmS is therefore expressed in all neuronal cell types but not in glial cell processes at this early stage of retinal development. Blue arrows in (**d**–**f**) point out that glial cells do not express SpmS. Note: there is no expression of SpmS in the inner plexiform layer (INL) that represents the forming synaptic area (GCL—Ganglion Cell Layer; IPL—Inner Plexiform Layer, the area where synapses are developing; NBL—Neuroblast Layer, a zone of cell proliferation of the inner optic cup that comprises progenitors. Scale = 20 µm).

**Figure 2 biomolecules-15-01374-f002:**
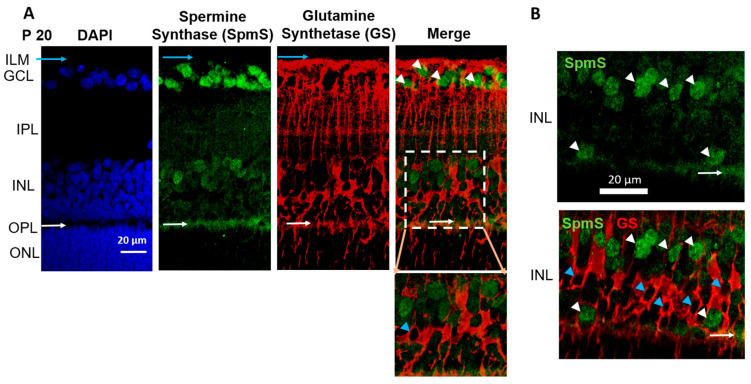
Localization of Spermine Synthase (SpmS) in P20 retinal neurons, but not in glial Müller cells. Spermine Synthase (SpmS) and Glutamine Synthetase (GS) immunolabeling in a 21-day-old (weanling: P20) rat retina. (**A**) SpmS (green) is found in neuronal cells: in ganglion cell somata and in neuronal cells in the inner nuclear layer, while GS (red) is localized in glial cells. White arrowheads point to nuclei of ganglion cells. Blue arrows point to the endfeet of Müller glia cells. Blue arrowhead points to the nucleus of a Müller cell. White arrows point to synapses in the outer plexiform layer. No co-localization was found between GS and SpmS. The insert illustrates Müller cells (red) without a co-localization with SpmS (green). (**B**) A random sample from the inner nuclear layer of the P20 rat retina shows no co-localization of SpmS and bipolar cells. White arrowheads point to nuclei of bipolar cells. Blue arrowheads point to the nuclei of Müller cells. White arrows point to synapses in the outer plexiform layer. (ILM-inner limiting membrane; GCL-ganglion cell layer; INL-inner nuclear layer; OPL-outer plexiform layer; ONL-outer nuclear layer, the bodies of rod and cone photoreceptors. Scale = 20 µm).

**Figure 3 biomolecules-15-01374-f003:**
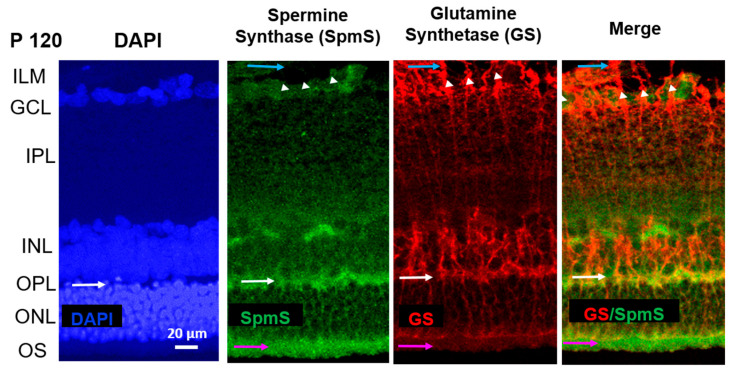
Localization of Spermine Synthase (SpmS) in P120 retinal aging synapses, not in glial Müller cells. Spermine Synthase (SpmS) and Glutamine Synthetase (GS) immunolabeling in the whole retina of a 120-day-old (P120) aging rat. White arrows indicate synapses in the outer plexiform layer. Blue arrows indicate the endfeet of Müller glial cells, white arrowheads indicate the nuclei of ganglion cells, and pink arrows indicate the outer segments of photoreceptor cells. SpmS (green) has been expressed in several ganglion cells and mainly in synapses, but not in bipolar cells or in the compartments of the Müller cells. Compared to the retina at P20 (Figure 2), the SpmS expression at P120 is less expressed in the neurons of older rats; however, it is strongly pronounced in synapses (white arrows). In the “Merge,” there is no co-localization of SPM synthesis (SpmS, green) and glial cells (GS, red). The SpmS label strongly highlights synapses between bipolar and photoreceptor cells in OPL (white arrows). Additionally, pink arrows highlight a strong SpmS label in the outer segments of photoreceptor cells. (ILM-inner limiting membrane; GCL-ganglion cell layer; IPL-inner plexiform synaptic layer; INL-inner nuclear layer; OPL-outer plexiform synaptic layer; ONL-outer nuclear layer formed by the rod and cone photoreceptors’ bodies; OS-outer segments of photoreceptors. Scale = 20 µm).

**Figure 4 biomolecules-15-01374-f004:**
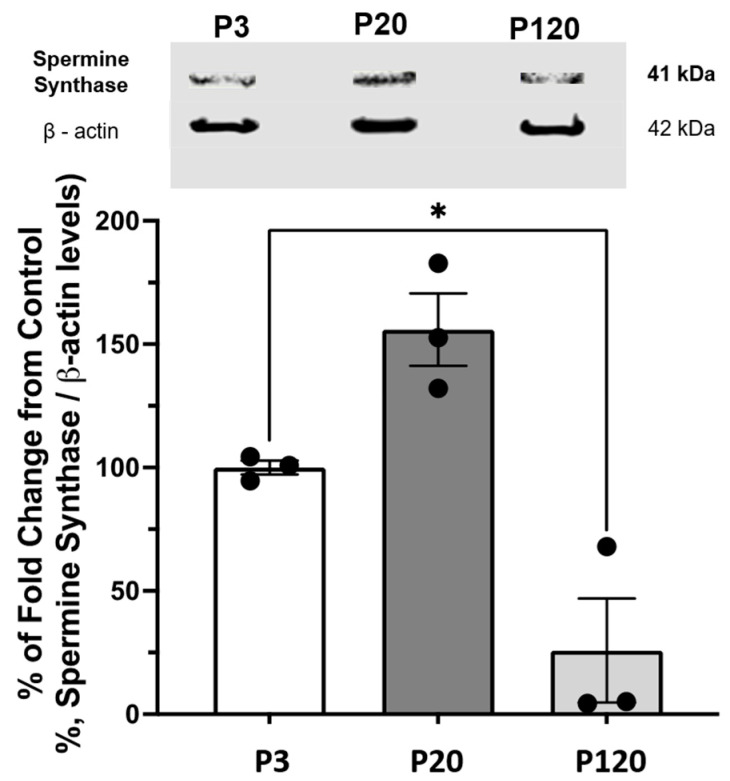
Western Blot of SpmS content in the whole retina during aging. The graph displays the quantification of the SpmS protein expression ± standard error of the mean (SEM) in whole retina lysates of 3 (P3), 21 (P20), and 120 (P120) days old rats. SpmS represents a band at 41 kDa, which is consistent with the predicted molecular weight of spermine synthase. The results of separate experiments using three different rats for each age are shown. The difference from control (P-3) to (P-20) and (P-120) is shown. β-Actin was used as a loading control to normalize the data. (One-way ANOVA results: *p* = 0.0224); values depicted are mean ± SEM; *n* = 3; Asterisk (*) represents statistical difference between P3 and P120. We found no statistical difference between P3 and P20.

**Figure 5 biomolecules-15-01374-f005:**
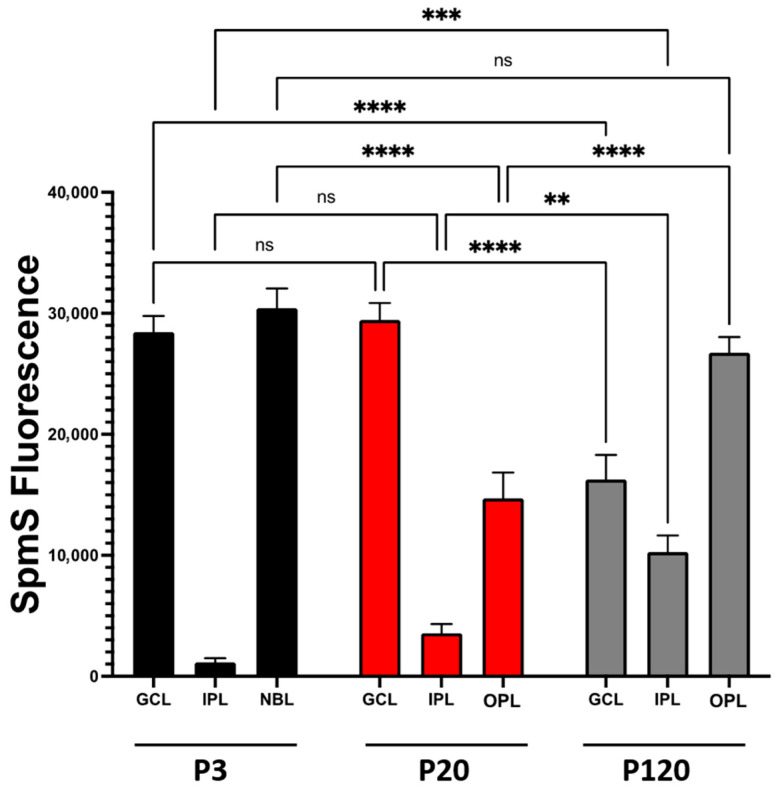
SpmS fluorescence redistribution between retinal layers during aging. ImageJ-Fiji NIH software was used for semi-quantitative analysis of SpmS fluorescence comparison between four retinal layers (ganglion cell layer (GCL), neuroblast (NBL), inner plexiform layer (IPL), and outer plexiform layer (OPL) from 3-, 21-, and 120-day-old rat retina. In the graph, the statistical difference within each group and between the groups is shown. In addition, there were statistical differences amongst groups: GCL (P20 vs. P120), IPL synaptic area (P20 vs. P120), and OPL (P20 vs. P120). Samples with an asterisk (**) indicate *p* values lower than 0.005, (***) *p* < 0.0005, (****) *p* < 0.00005 with a 95% confidence interval, which was considered statistically different. (ns) indicate no statistical difference among the samples. The data clearly show (i) the drop of SpmS signal in ganglion cells from P3-P20 down to P120, (ii) a robust increase in the SpmS activity in synapses of ganglion cells (IPL) during maturation and in adulthood, and (iii) strong growth of SpmS activity in synapses of photoreceptors (OPL). The shift in SpmS label from ganglion cells at an early age to the synapses of amacrine, ganglion, and bipolar cells (in IPL) and to a very high activity of SpmS in the synapses of photoreceptors (OPL) in adults. We measured 7 different areas within each layer in *n* = 3 experiments.

**Figure 6 biomolecules-15-01374-f006:**
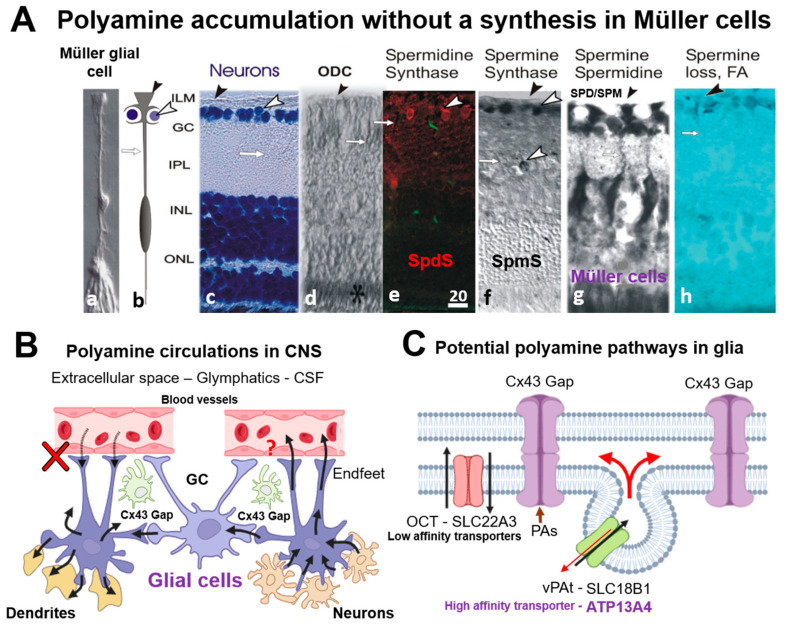
Localization of Spermidine synthase (SpdS), Spermine Synthase (SpmS), and spermine (SPM), spermidine (SPD) in adult retina (**A**), suggesting polyamine exchange between neurons and glia (**B**), and potential pathways for SPD/SPM in glial cells (**C**). (**A**)—The illustration of Müller cells and neurons in retina with polyamine-synthesizing enzymes and their products. a—Isolated Müller cell [96], b—a scheme of Müller cell enwrapping the neurons, ganglion cells, by the endfoot processes. Black arrowhead indicates the endfeet of Müller glial cells, and white arrowhead points to ganglion cell, c—Neurons stained with cresyl violet (blue labeling). White arrowhead shows ganglion cells (GC), white arrow points to a stalk of Müller glial cell, and black arrowhead shows the vitreal surface of the retina where glial cell endfeet form the inner limiting membrane, ILM. (Abbreviations: GC-ganglion cell layer; IPL-inner plexiform synaptic layer; INL-inner nuclear layer, comprising the neuronal bodies of bipolar and horizontal cells and Müller cells; ONL-outer nuclear layer, containing the cell nuclei of rod and cone photoreceptors); d—Ornithine decarboxylase (ODC) immunolabeling (black asterisk). ODC is the enzyme synthesizing putrescine, the primary polyamine from which spermidine is produced by SpdS. ODC is absent in glial cells, but may be present in photoreceptor cells [7]. Black arrowhead points to ILM; white arrows point to a stalk of a glial cell. e—SpdS immunolabeling (red label) in the adult rat retina [51]. White arrowhead indicates the ganglion cells’ bodies stained with anti-SpdS antibody. White arrow indicates Müller cell’s stalk. (Note: green fluorescence is an autofluorescence from blood vessels) f—SpmS (black label) has been localized in ganglion cells, and in some bipolar cells, but not in Müller cells (DAB-nickel contrasting) [18]. The SpmS expression is not pronounced in all neurons but in a few (white arrowheads). g—SPD/SPM immunolabels show that only Müller cells accumulate these polyamines, but not neurons [8]. h—Gliotoxin, mono-fluoroacetate (FA) depletes SPD/SPM from Müller cells [47]. (Scale = 20 µm; Müller cells span from the border between the retina and vitreal humor). (**B**)—Suggested scheme of polyamine circulation in CNS. SPD/SPM are taken up by glial cells from the neurons and redistributed among other glial cells (astrocytes or Müller cells) via gap junctions (Cx43 Gap) and synapses. There is no uptake from blood vessels (red cross [13]), and the release of PAs to blood vessels is not known (red question sign). (C)—Potential pathways for PAs to be transported into glial cells and between glial cells. Membrane organic cation transporters (OCT) such as SLC22A3 and SLC18B can be involved; however, it is a low-affinity transporter [54,55], while a high-affinity transporter, ATP13A4, may be a good candidate [93]. Glial cells may exchange PAs in their syncitium via connexin gap junctions (Cx43 Gap [43,44,57,63,67].

## Data Availability

Additional data supporting the conclusions of this finding will be made available without undue reservation.

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
