# Peer review of "Age-Dependent Redistribution of the Life-Important Enzyme in the Retina: Adult Müller Glial Cells’ Endfeet Lack Spermine Synthase Expression"

_biomolecules, 2025, doi:10.3390/biom15101374_

Round 1
Reviewer 1 Report
Comments and Suggestions for Authors
My four comments are shown below.
- The discussion is not supported by the present data, because authors performed expression and localization studies on SpmS only.
- Localization studies on SpmS seems to be fine. However, they are not enough to suggest the significant roles of PA and their enzymes in the retina, and more analyses and results will be essential.
- Authors studied with rats, and it is unclear how P3, P20 and P120 are expected to be comparable to human age.
- Introduction and Discussion sections fail in explaining the significance of study localization of SpmS in the retina, and it will be difficult for readers to understand the relationship between PA and the retinal diseases or health.
Author Response
Please, see the attached document

Reviewer 2 Report
Comments and Suggestions for Authors
The contribution submitted by Zayas-Santiago describes the importances of Spermine synthase expression during development and makes special emphasis it's lack of expression in Muller glia, taking into consideration the spermine localization in these radial glia cells. I tis a well-written easy to follow contribution that would benefit of minor changes before it is published.
- The authors might consider to present a figure in which SpmS expression in tissue sections is presented as a function of developmental stage. The reading and comparing will be far easier.
- Figure 4 has to be improved. The representative blot does not meet (under my criterion) publication quality. Both bands have to be presented in the same blot, and to do so the extracts have to analyzed in either a 20% gel or a gradient gel. Even if the authors are no willing to do that, another blot should be mounted and shown, Better, all the blots should be uploaded for reviewing.
- A final figure summarizing their results in the form of a cartoon is desirable.
Author Response
Comments and suggestions for authors:
The contribution submitted by Zayas-Santiago describes the importances of Spermine synthase expression during development and makes special emphasis it's lack of expression in Muller glia, taking into consideration the spermine localization in these radial glia cells. I t is a well-written easy to follow contribution that would benefit of minor changes before it is published.
We thank the reviewer for recognizing that we presented a well-written and easy-to-follow contribution.
- The authors might consider to present a figure in which SpmS expression in tissue sections is presented as a function of developmental stage. The reading and comparing will be far easier.
This is a great suggestion, and we appreciate the reviewer. As shown in Figure 5, such an idea has already been presented. We demonstrated an increase in SpmS signal in the synaptic layer, IPL. Later, in our planned review on the polyamine exchange mystery, we can create a similar graph highlighted by a rising trend. We also plan to include future review studies on both synthases, SpdS and SpmS.
- Figure 4 has to be improved. The representative blot does not meet (under my criterion) publication quality. Both bands have to be presented in the same blot, and to do so the extracts have to analyzed in either a 20% gel or a gradient gel. Even if the authors are no willing to do that, another blot should be mounted and shown, Better, all the blots should be uploaded for reviewing.
The Western blot (Fig. 4) is presented as a sample, while detailed quantitative analyses were carried out using NIH-developed software to measure SpmS expression fluorescence. This provides a significant advantage because we compare not only the total amount of SpmS expression, as shown in the simple Western blot, but also perform layer-by-layer analyses of the retinas (Fig. 5). We appreciate the Reviewer's question; however, we lack the resources to repeat this complex research entirely. We would need to order (i) new animals at three different ages and (ii) purchase a new antibody to replicate the entire study. In addition, three of the co-authors have recently transitioned to new professional positions.
- A final figure summarizing their results in the form of a cartoon is desirable.
This is an excellent suggestion and we created a cartoon summarizing the study.
Reviewer 3 Report
Comments and Suggestions for Authors
The manuscript titled “Age-dependent redistribution of the life-important enzyme in retina: Adult Müller glial cells’ endfeet lack spermine synthase expression,” by Astrid Zayas-Santiago and colleagues, investigates the developmental stage-dependent expression and localization of spermine synthase (Sms/SpmS) in Müller glial cells of the retina, highlighting its implications for neuronal function and central nervous system diseases. Their findings suggest that, like spermidine, Müller glial cells do not synthesize spermine but may accumulate it from extracellular sources.
Immunofluorescence of rat retinal tissue slices at postnatal day 3 revealed that SpmS was widely expressed in neuroblast cells but absent in Müller cells (identified by glutamine synthetase (GS) immunoreactivity) and synaptic layers. By postnatal day 21, SpmS was strongly expressed in neuronal cells, particularly in ganglion and bipolar cells, but not in glial cells. And at postnatal day 120, SpmS was found in synaptic areas, with significantly reduced presence in neuronal soma and minimal detection in Müller cells. Western blot analysis of total retinal tissue indicated a decrease in SpmS expression during maturation, most notably between postnatal days 20 and 120. As polyamines (PAs) play a crucial role in regulating various receptors and channels in the central nervous system (CNS), understanding the localization of thier synthesis and storage is essential for elucidating their roles in a growing number of neurodegenerative conditions as well as aging.
The manuscript is very well written, including an informative introduction that effectively lays the groundwork supporting the relevance of the study. The immunofluorescence images of the retina are beautiful, and the data have potential to broadly impact the field. Because of this potential importance, I am concerned that this antibody has not been validated to ensure selectivity to SpmS. In particular, as it seems that spermidine synthase may have a similar distribution pattern, it is imperative that this occurs. It does not appear that the vendor has validated the antibody in its current formulation. Additionally, the Western blot image appears to have been manipulated from the original, which also includes additional bands that could be indicative of SpdS. A cleaner image should be provided for figure 4.
Some minor suggestions:
Line 74: “rate” – should this be “conversion rate” or “ratio”?
Lines 105-107 suggest that Spm must be synthesized within the CNS. It sounds like the same might be true for Spd, and yet Spd supplementation appears to be beneficial. It would be helpful to clarify whether, like Spd, supplementation with Spm could support glial cells.
Section 2.4: mention what tissues are used for Western blot
Throughout: abbreviations only need to be defined at first use
Line 420 and in Figure legend 5: refers to SpmS “activity”. “signal” would be more appropriate.
Line 450: misplaced “(i)”
Discussion page 12 last paragraph: how do these findings differ from what is known about SpdS in these models?
Author Response
The manuscript titled “Age-dependent redistribution of the life-important enzyme in retina: Adult Müller glial cells’ endfeet lack spermine synthase expression,” by Astrid Zayas-Santiago and colleagues, investigates the developmental stage-dependent expression and localization of spermine synthase (Sms/SpmS) in Müller glial cells of the retina, highlighting its implications for neuronal function and central nervous system diseases. Their findings suggest that, like spermidine, Müller glial cells do not synthesize spermine but may accumulate it from extracellular sources.
Immunofluorescence of rat retinal tissue slices at postnatal day 3 revealed that SpmS was widely expressed in neuroblast cells but absent in Müller cells (identified by glutamine synthetase (GS) immunoreactivity) and synaptic layers. By postnatal day 21, SpmS was strongly expressed in neuronal cells, particularly in ganglion and bipolar cells, but not in glial cells. And at postnatal day 120, SpmS was found in synaptic areas, with significantly reduced presence in neuronal soma and minimal detection in Müller cells. Western blot analysis of total retinal tissue indicated a decrease in SpmS expression during maturation, most notably between postnatal days 20 and 120. As polyamines (PAs) play a crucial role in regulating various receptors and channels in the central nervous system (CNS), understanding the localization of thier synthesis and storage is essential for elucidating their roles in a growing number of neurodegenerative conditions as well as aging.
The manuscript is very well written, including an informative introduction that effectively lays the groundwork supporting the relevance of the study. The immunofluorescence images of the retina are beautiful, and the data have potential to broadly impact the field.
We appreciate your highly positive evaluation of our manuscript as very well written, including an informative introduction that effectively lays the groundwork supporting the relevance of the study. The immunofluorescence images of the retina are beautiful, and the data have potential to broadly impact the field. We thank the reviewer, and we are looking forward to writing a review where we can combine studies on both SpdS and SpmS for broad readers.
Because of this potential importance, I am concerned that this antibody has not been validated to ensure selectivity to SpmS. In particular, as it seems that spermidine synthase may have a similar distribution pattern, it is imperative that this occurs. It does not appear that the vendor has validated the antibody in its current formulation.
According to a vendor (Abcam), the anti-SpmS antibody (Abcam, rabbit, monoclonal, ab248996, 41 kDa) does not cross-react with the anti-spermidine synthase antibody (SpdS). Previously, we used the anti-SpdS antibody (Abcam, rabbit, polyclonal, ab111884, 34 kDa), which also does not cross-react with SpmS. (We added this paragraph to the Methods section).
Furthermore, comparing the labeling for SpdS (Zayas-Santiago et al., 2024) with the current SpmS labeling reveals a different pattern than what SpmS shows, especially in the synaptic inner plexiform layer (IPL) and notably in the outer plexiform layer (OPL). As shown earlier, SpdS does not label the synapses in the OPL. Here, we find that SpmS exhibits intense staining in these synapses of adult and mature retinas (Fig. 3, P120, yellow horizontal arrows).
Additionally, the Western blot image appears to have been manipulated from the original, which also includes additional bands that could be indicative of SpdS. A cleaner image should be provided for figure 4.
We collected several retinal tissue samples (without pigment epithelium) for Western blot analysis to statistically evaluate SpmS changes during aging. Β-Actin standards were used as a loading control to normalize WB data.
Some minor suggestions:
Line 74: “rate” – should this be “conversion rate” or “ratio”?
Thank you. We corrected “rate” to “conversion rate”
Lines 105-107 suggest that Spm must be synthesized within the CNS. It sounds like the same might be true for Spd, and yet Spd supplementation appears to be beneficial. It would be helpful to clarify whether, like Spd, supplementation with Spm could support glial cells.
Section 2.4: mention what tissues are used for Western blot
We collected several retinal tissue samples (without pigment epithelium) for Western blot analysis to statistically evaluate SpmS changes during aging. Β-Actin standards were used as a loading control to normalize WB data. Also, since our students who moved to study medicine are in different countries, we have limited access to original data to demonstrate India ink. Of course, with more time, we can find the original data or redo the experiments.
Throughout: abbreviations only need to be defined at first use
We corrected it in the manuscript, thank you.
Line 420 and in Figure legend 5: refers to SpmS “activity”. “signal” would be more appropriate.
We replaced activity for signal in the text.
Line 450: misplaced “(i)”
We deleted the misplaced (i) changing to numbers
Discussion page 12 last paragraph: how do these findings differ from what is known about SpdS in these models?
We made modifications to the discussion and included a new image (fig 6) talking about what is known about SpdS in a link with SpmS and polyamine exchange in the retina and CNS (made a common scheme of PA turnover in the different models to clear up, simplify, and facilitate the information to the reader.
Reviewer 4 Report
Comments and Suggestions for Authors
The manuscript from Zayas-Santiago and colleagues analyses the expression of spermine synthase in the rat retinas from post natal day 3 to post natal day 120. The manuscript is based on some Immunohistochemistry analysis of Spermine synthase, glutamine synthetase as marker of glial cells, and DAPI as reference for the cell nuclei. Spermine synthase is a key enzyme involved in the synthesis of the polyamine spermine, and thus this manuscript falls within the frame of polyamine roles in brain development and physiology, a very interesting and studied topic.
The manuscript is clear and easy to understand. Here my specific comments:
- Even if the glutamine synthetase is used as marker of glial cells, the addition of an Immunohistochemistry with a marker of neurons would be very important to clearly address spermine synthase (SpmS) localization, in particular at day 20 and 120 when neuronal differentiation already occurs.
- The authors showed a western blot analysis of the whole retina at the three time points. To strengthen this data, the authors should show the mRNA expression by qPCR of the spermine synthase in the retina at the three time points.
- In figure 5, the authors showed Semi-quantitative analysis of SpmS redistribution between retinal layers and during aging based on SpmS fluorescence comparison. It is not clear how the fluorescence has been normalized between the experiments.
Minor points:
- Please avoid the excessive use of bullets (i), (ii) … in the text. For example: rows 27-28 page 1, rows 52-53 page 2
- Row 74 page 2 “rate” should be “ratio”
- The reviewer is a little concerned by the use of the word “aging” to refer to 120d old rat. Should 120 day old rat be considered just adult?
- In the discussion section, page 13 lane 503-506: “Notably, when PA synthesis was inhibited in cultured astrocytes, the cells did not survive or proliferate. However, supplementing with spermidine (SPD) restored both survival and proliferation [26]. This indicates that glial cells depend on the external uptake of PAs, and the protective effect of SPD on developing glial cells was negated when PA transport was hindered using a specific transporter inhibitor [26]” In my opinion, the fact that exogenous spermidine supplementation restores survival and proliferation of glial cells indicates that spermidine is essential to survival of glial cells and that glial cells are able to efficiently import spermidine, not that “glial cells depend on the external uptake of Pas” otherwise the inhibition of the PA synthesis would not have an effect. Please correct and improve this discussion section.
Author Response
The manuscript from Zayas-Santiago and colleagues analyses the expression of spermine synthase in the rat retinas from post natal day 3 to post natal day 120. The manuscript is based on some Immunohistochemistry analysis of Spermine synthase, glutamine synthetase as marker of glial cells, and DAPI as reference for the cell nuclei. Spermine synthase is a key enzyme involved in the synthesis of the polyamine spermine, and thus this manuscript falls within the frame of polyamine roles in brain development and physiology, a very interesting and studied topic.
We sincerely appreciate that our manuscript is “a very interesting and studied topic.” We have made efforts to ensure the manuscript is clear and easy to understand.
The manuscript is clear and easy to understand.
We enjoy that the reviewer likes the clarity of the manuscript and easy to follow text.
Here my specific comments::
- Even if the glutamine synthetase is used as marker of glial cells, the addition of an Immunohistochemistry with a marker of neurons would be very important to clearly address spermine synthase (SpmS) localization, in particular at day 20 and 120 when neuronal differentiation already occurs.
Retinal neurons exhibit different markers during development, making it unreliable to depend solely on transient neuronal markers. However, (i) retinal layers are clearly defined and well understood, and (ii) neurons are consistently stained with DAPI regardless of age. Thus, the retina is simply organized into layers, with neuronal layers distinctly separated. The only Müller cells extend throughout the retina, spanning from the inner limiting membrane (ILM) to the outer segments (OS) of photoreceptors, which is clearly visible when using Glutamine synthase as a specific, age-independent glial marker. We did not use the common GFAP labeling, as it may be absent in healthy glia because it is a marker of reactive glial cells.
Since all images were taken at the same time using the same software and parameters in confocal microscopy, their intensities were normalized. After this normalization, the images were processed with a program that scans retinal layers and calculates data for plotting on graphs. Minor points: previous research has shown that the SPM precursor, SPD, is produced by spermidine synthase (SpdS) in adult neurons of the rat retina marked by DAPI, but not in glial cells. To repeat this study again using different markers, we would need to order (i) new animals at three different ages and (ii) purchase a new antibody to replicate the entire study. In addition, three of the co-authors have recently transitioned to new professional positions.
- The authors showed a western blot analysis of the whole retina at the three time points. To strengthen this data, the authors should show the mRNA expression by qPCR of the spermine synthase in the retina at the three time points.
Unfortunately, we are currently unable to perform the suggested mRNA analysis because we would need to order (i) new animals at three different ages, (ii) purchase new kits, and (iii) since the funding for the research has ended, three of the co-authors have recently moved on to new professional positions.
- In figure 5, the authors showed Semi-quantitative analysis of SpmS redistribution between retinal layers and during aging based on SpmS fluorescence comparison. It is not clear how the fluorescence has been normalized between the experiments.
The NIH-created method is widely used as a quantitative method when the images are normalized. Since all images were taken in one time and with the same set of software programing at the confocal microscopy, the intensities of all images were thus normalized. After such normalization the images were processed via the program that scans retinal layers and calculates numerically with following plotting to graphs.
Minor points:
- Please avoid the excessive use of bullets (i), (ii) … in the text. For example: rows 27-28 page 1, rows 52-53 page 2
We deleted some that were repetitive.
- Row 74 page 2 “rate” should be “ratio”
We deleted rate and included ration, thanks for the recommendation.
- The reviewer is a little concerned by the use of the word “aging” to refer to 120d old rat. Should 120 day old rat be considered just adult?
- In the discussion section, page 13 lane 503-506: “Notably, when PA synthesis was inhibited in cultured astrocytes, the cells did not survive or proliferate. However, supplementing with spermidine (SPD) restored both survival and proliferation [26]. This indicates that glial cells depend on the external uptake of PAs, and the protective effect of SPD on developing glial cells was negated when PA transport was hindered using a specific transporter inhibitor [26]” In my opinion, the fact that exogenous spermidine supplementation restores survival and proliferation of glial cells indicates that spermidine is essential to survival of glial cells and that glial cells are able to efficiently import spermidine, not that “glial cells depend on the external uptake of Pas” otherwise the inhibition of the PA synthesis would not have an effect. Please correct and improve this discussion section.
This is an excellent question. As Kang et al. (2011) state in Nature (478(7370):483-9. doi: 10.1038/nature10523), CNS maturation—where glial cell genes play a key role—is completed in humans between 2 and 6 years after birth. In rodents, Kir4.1 is not present during fetal live and earlier neonatal period, but is developed roughly during 11 to 20 postnatal days [ref.20]. After this period, the healthy CNS remains mostly unchanged in gene expression in both humans and rodents. We observed that the KCNJ10 gene, which encodes the polyamine-sensitive potassium channel Kir4.1—the primary glial cell channel encoded by the gene—is fully functional by P20-P120, and we plotted the development graph (Olsen et al., 2015 [20]). Therefore, P3 in rodents roughly corresponds to 6 months in humans, P20 in rodents corresponds to 4-6 years in humans, and P120 indicates a fully mature adult.
According to the Reviewer’s suggestion, we placed the following paragraph in the Discussion section:
“The KCNJ10 gene, which encodes the potassium channel Kir4.1 (the major channel found in glial cells) are expressed between postnatal days 11 and 20, while in humans, they are expressed from approximately 2 to 6 years after birth [20]. After this period, gene expression in the healthy central nervous system (CNS) remains relatively stable in both humans and rodents [20]. Intriguingly, we found that spermine synthase (SpmS) is not expressed at this age.
In rodents, P3 approximately corresponds to 6 months in humans, while P20 equates to about 4 to 6 years in humans [20]. The P120 pertains to a mature adult. Thus, the accumulation of SPM in glial cells is not the result of their synthesis but rather due to transport mechanisms.”
.
Round 2
Reviewer 1 Report
Comments and Suggestions for Authors
N/A